# Modulatory Effect of Gut Microbiota on the Gut-Brain, Gut-Bone Axes, and the Impact of Cannabinoids

**DOI:** 10.3390/metabo12121247

**Published:** 2022-12-10

**Authors:** Iddrisu Ibrahim, Soumyakrishnan Syamala, Joseph Atia Ayariga, Junhuan Xu, Boakai K. Robertson, Sreepriya Meenakshisundaram, Olufemi S. Ajayi

**Affiliations:** 1The Microbiology Program, Department of Biological Sciences, College of Science, Technology, Engineering, and Mathematics (C-STEM), Alabama State University, Montgomery, AL 36104, USA; 2Departments of Medicine, SUNY Downstate Health Sciences University, Brooklyn, NY 11203, USA; 3The Industrial Hemp Program, College of Science, Technology, Engineering, and Mathematics (C-STEM), Alabama State University, Montgomery, AL 36104, USA; 4Department of Microbiology and Biotechnology, JB Campus, Bangalore University, Bangalore 560 056, Karnataka, India

**Keywords:** gut-bone axis, gut-brain axis, microbiome, probiotics, osteoporosis

## Abstract

The gut microbiome is a collection of microorganisms and parasites in the gastrointestinal tract. Many factors can affect this community’s composition, such as age, sex, diet, medications, and environmental triggers. The relationship between the human host and the gut microbiota is crucial for the organism’s survival and development, whereas the disruption of this relationship can lead to various inflammatory diseases. Cannabidiol (CBD) and tetrahydrocannabinol (THC) are used to treat muscle spasticity associated with multiple sclerosis. It is now clear that these compounds also benefit patients with neuroinflammation. CBD and THC are used in the treatment of inflammation. The gut is a significant source of nutrients, including vitamins B and K, which are gut microbiota products. While these vitamins play a crucial role in brain and bone development and function, the influence of gut microbiota on the gut-brain and gut-bone axes extends further and continues to receive increasing scientific scrutiny. The gut microbiota has been demonstrated to be vital for optimal brain functions and stress suppression. Additionally, several studies have revealed the role of gut microbiota in developing and maintaining skeletal integrity and bone mineral density. It can also influence the development and maintenance of bone matrix. The presence of the gut microbiota can influence the actions of specific T regulatory cells, which can lead to the development of bone formation and proliferation. In addition, its metabolites can prevent bone loss. The gut microbiota can help maintain the bone’s equilibrium and prevent the development of metabolic diseases, such as osteoporosis. In this review, the dual functions gut microbiota plays in regulating the gut-bone axis and gut-brain axis and the impact of CBD on these roles are discussed.

## 1. Introduction

The gut contains diverse microbial flora, including bacteria, fungi, archaea, protozoa, and viruses. Most gut microbes live in harmony with the human host, interact constantly and regularly with the human body and modulate various vital body activities [1]. The gut contains over a thousand bacterial species, including over 50 genera. In the gastrointestinal tract, the large intestine alone possesses an incredibly high number of bacteria between 10^11^–10^12^ [2,3]. A deeper understanding of the microbiome has shaped the entire domain of disease pathogenesis and helped increase potential prevention strategies. A healthy and diverse microbial ecosystem of the gut is vital in maintaining optimal health, and yet these same microbes can instigate a series of debilitating diseases, especially in genetically predisposed people when disturbed [4]. The importance of the gut microbiome also lies in its ability to influence the hosts’ metabolism through varied inherent pathways, shaping and modulating immunity. It is, therefore, imperative to keep a steady and balanced microbiota [4,5].

The advent of the “Human Genome Project” revitalized our realization and appreciation that the human body has a rich ecosystem of microorganisms. Before the “Human Genome Project,” our understanding was limited to only microorganisms that can be cultured in the laboratory (i.e., the 5% culturable ones). In the space of ten years, molecular tools have been designed to enable us to not only understand the microbiome, but also, the “parasitome,” the “virome,” and the “fungome” [6].

These molecular tools has helped us uncover several diverse belligerent microorganisms that either negatively or positively influence our health. The recent outbreak of the SARS-CoV virus, which caused the COVID-19 pandemic and the reemergence of monkeypox, renders some pungent examples [7]. The more we delve deeper into understanding the inter-relationship between the microbiome and ourselves, the better it will be for us to draw closer to predicting disease pathogenesis, curating mitigation strategies, and also harnessing and enriching the “good guys” of the microbiome [8,9].

Medicinal cannabis is increasingly becoming popular, partly due to the realization of its impact on psychiatric and neurological disorders. Cannabinoids are molecules of pharmaceutical relevance, and they influence the treatment of many ailments. Regardless of the many potential impacts of the plant, the prescription of medicinal cannabis is highly conserved [10]. Several published critical pieces of research on medicinal cannabis have highlighted its importance and thus helped minimize the social stigma that causes many jurisprudences to frown on the plant. Several states of USA, Israel, Canada, Uruguay, and Australia are among some of the jurisdictions that lower their guide by paving ways to harness the plant’s potential [4]. This latest development excites many biotechnologies, farming organizations, and pharmaceutical cooperation, in the end, pushing them into healthy competition regarding the development of medical cannabinoids and their commercialization.

Delta-9-tetrahydrocannabinol (Δ-9 THC) is the only known variant among the 126 different variants identified to be psychoactive, potent, and affects brain development. This has become a worry to many clinicians because of its ability to induce mood, behavior cognition, and appetite [1]. Clinicians have also identified severe psychological and behavioral aberrations among chronic users of cannabis. The aberrations vary over time and individuality, rendering the treatment of long-term effects of cannabis complex and unpredictable with current technologies. The precariousness of ∆-9 THC has moved the attention of contemporary clinicians and researchers to venture into another variant known as CBD. CBD has already shown many of its therapeutic potential among the cannabinoids, and it is also abundant in the cannabis plant, its profile is demonstrated to possess unique health benefits, and lacks any form of psychoactive effects [11]. However, THC and CBD have relative synergy; THC can reinforce the beneficial profile of CBD, and CBD, on the other hand, can curtail the euphoric and psychoactive profile of THC [12]. CBDs’ mechanisms of action are poorly understood, and it does not interact with similar cannabinoid receptors as THC. CBD mode of action is attributed to the inhibition of anandamide degradation and anti-inflammatory, serotoninergic and antibiotic profiles [13]. CBD shows strong pharmacodynamics and pharmacokinetic properties that could result in adverse drug events and drug–drug interaction. CBD may be administered as a purified product, consumed as a whole cannabis/hemp plant component, and as an extract from the cannabis plant. Several biological targets of drugs are affected by CBD, thus increasing the possibilities of CBD-drug interactions during medications. For instance, CBD is known to affect the CYP3A4/2C19 which is critical in drug metabolism, additionally, CBD is implicated in drug excretion too, affecting most P-glycoproteins [10,14]. Cytochrome P450 enzymes metabolizes and are also involved in the biotransformation of CBD [15]. Sedation and somnolence are potential biological effects of cannabis potentiated by medications of like effects such as opioids. CBD and other cannabinoids further have their inherent biological targets that may certify as adverse drug events autonomous of any form of drug–drug interaction capabilities [4].

The human host is able to provide nutrients and other resources to support the development of the life of microbes by supplying them with secondary metabolites. The gut microbiota plays a vital role in the regulation of both innate and acquired immunity. It also produces immunological intermediates, which are essential in the development of anti-inflammatory and proinflammatory responses. The interaction between the gut and other distant organs is evidenced by the presence of several essential loops [16,17,18,19]. These include gut-brain, gut-bone, gut-lung, and gut-heart. The presence of altered microbial diversity in the gut can contribute to the development of various diseases. In addition, the interactions between the different intermediates of the gut microbiome and other organs can cause complex changes that can lead to the development of diseases such as cancer, inflammatory bowel disease, diabetes, neurodegenerative and autoimmune disease. Understanding the interactions between the various components of the human body’s microbiome and other distant organs is very important in the treatment of diseases, especially those that have no cure [20,21,22,23,24].

The process of bone remodeling is carried out throughout one’s life. It involves replacing old and worn-out bones with new ones that are healthy and also the process of repairing micro-damages. It also regulates the balance of calcium homeostasis. The complex actions of the bone resorbers and osteoblasts are responsible for the development of new healthy bones [25]. The delicate equilibrium between the two dominant types of bone cells, osteoblasts, and osteoclasts, is responsible for the development and maintenance of bones. Any disruption or alteration to this equilibrium can lead to diseases that affect the skeletal system [26]. The regulation of bone homeostasis involves the intercellular cross-talk between chondrocyte, osteoblast, osteoclast, and osteocyte. Complex cell signaling networks are responsible for the coordination of this process. In recent years, the link between the metabolism of bone and the gastrointestinal tract has been established [27].

The development and progression of osteoporosis are characterized by the reduction of bone mass and the degradation of its micro-architectural properties. This condition can lead to the development of fractures. The prevalence of osteoporosis has drastically changed in young people, especially those active in their early years [28]. The most common type of this disease is postmenopausal osteoporosis. It is caused by estrogen deficiency. Although estrogen replacement therapy is considered a standard treatment for postmenopausal osteoporosis, it can also increase the risk of developing breast and other types of cancer. Some of the other options that are commonly used for treating this condition include the use of a combination of drugs known as bisphosphonates and teriparatide [29,30,31,32,33,34,35,36]. Treatment for osteoporosis is very challenging. The various factors that affect a person’s chances of developing osteoporosis are also considered when choosing the appropriate anti-osteoporotic therapy. These include the type of cancer, age, gastrointestinal issues, and environmental triggers. In addition to these, other factors such as genetic predisposition and the consumption habits of individuals can also affect a person’s chances of developing this condition [37,38,39,40]. Additionally, dietary supplements can help prevent or manage osteoporosis, and they can also be very helpful in controlling the development and progression of this condition [39,40,41,42,43]. The importance of the gut microbiome has been acknowledged as a potential tool for treating various human diseases. The complex cross-talk of the gut-bone axis can help determine the outcome of skeletal health and osteointegration [37,38,39,40,41,42,43]. Through studies on the interactions between the gut microbiome and bone metabolism, researchers have gained a deeper understanding of the mechanisms that can affect the development and maintenance of osteoporosis. This review seeks to elucidate the functions gut microbiota play in regulating gut-bone and gut-brain axes and the impact of CBD on these roles in maintaining a healthy gut microbiome.

### 1.1. The Gut Microbiota

The term “gut microbiota” refers to the microorganisms (e.g., bacteria, archaea, fungi, and viruses) that live in the digestive tract of animals. There are various terms used to refer to this type of flora, e.g., “gut flora” [44]. The gastrointestinal metagenome, which is the aggregate of all the genome sequences of gut microbiota, is also sometimes referred to as the microbiome [45]. The human gut is the primary location of the microbiota. Its diverse bacterial communities play a variety of roles in the development and maintenance of human health [46]. Some of these include the control of immune function, the maintenance of the intestinal epithelium, and the metabolism of pharmaceutical and dietary compounds. Depending on the region of the digestive tract, the composition of the gut microbiota can vary. The highest density of microbial communities can be found in the colon, which has around 300 to 1000 species [47]. The majority of the bacteria in the gut come from around 30 to 40 species. Around 60% of the feces’ dry mass is composed of bacteria. Some of the bacteria in the gut are aerobic, while the other types are anaerobes [48], however, depending on the location in the gut, several different enzymatic and chemical assays can be used to identify and differentiate bacteria in the different parts of the gut. For instance, using cultured strain of *Bacteroides fragilis* and intestinal *Bacteroides*, various experimental assays were used to differentiate the *Bacteroides* based on the location of the bacteria. Pentose fermentation, acid end products, 10% bile tolerance and gas production was used as differentiation basis for separating the oral strains of *Bacteroides* (e.g., *Bacteroides oralis*) from the intestinal strains. The researchers noted that the oral strains produced succinic, lactic, acetic, and formic acids but no gas, and were killed by 10% bile, and could not ferment xylose and arabinose. The intestinal strains however produced succinic, formic, acetic, and propionic acids and gas, with a few strains forming lactic acid. These strains could be stimulated by 10% beef bile, and were demonstrated to ferment xylose and arabinose (e.g., *Bacteroides fragilis*) [49,50].

There are majorly five phyla that dominate the intestinal microbiota. These includes *Bacteroidota, Bacillota (Firmicutes), Actinomycetota, Pseudomonadota,* and *Verrucomicrobiota*. The *Bacteroidota* and *Bacillota* makes up 90% of the bacterial population in the gut [51]. In Table 1 the major Gut Microbiota found in the colon and their unique characteristics are listed.

### 1.2. Bacteria Inhabiting the Stomach

The stomach contains high acidity and thus most microorganisms are destroyed by the acidic environment. Due to the high acidity of the stomach, the few bacteria that thrives in the stomach include: *Streptococcus, Staphylococcus, Lactobacillus, Peptostreptococcus* and *Helicobacter pylori* [67]. The *H. pylori* is a known cause of gastric cancer and stomach ulcer [67].

### 1.3. Bacteria Inhabiting the Ileum

Most of the microorganisms that are commonly found in the stomach are rod-shaped and Gram-positive cocci. Due to the influence of the stomach on the development and maintenance of microorganisms in this area, there is a trace amount of these organisms in the upper chamber of the small intestines [67]. However, in the lower portion of the intestine, the conditions support the development and maintenance of Gram-negative bacteria. The small intestine’s bacterial flora plays a vital role in a wide range of functions. It provides a variety of regulatory signals that help the development and utility of the intestine. Overgrowth of bacteria can lead to intestinal failure [67,68].

### 1.4. Bacteria Inhabiting the Colon

The colon has a large bacterial population [69]. Almost all of the flora and large intestine’s obligate anaerobes are composed of *Bifidobacterium* and *Bacteroides* [70]. Some factors that can affect the growth and maintenance of the large intestine’s microorganism population include stress, parasites, and antibiotics [71].

## 2. Microbiome and Human Body Function

### 2.1. Microbial Components and Their Effect on Host Physiology

Some of the most common components of gut microbes are present on the cell walls and outer membranes. These molecules are known to interact with host tissues and are often the main products of the gut. For instance, peptidoglycan, a universal component of bacterial membranes, activates various immune signaling pathways [72]. More so, a significant component of the cell wall of Gram-negative bacteria is the LPS, which is known to be a potent systemic immune activator. Additionally, flagellins, which are widely expressed across different bacterial species, are immunostimulatory. Aside from stimulating the development of immune tolerance, these molecules can also help balance the populations of immune cells in the gut. For instance, the capsular polysaccharide, which is found in the gastrointestinal tract’s common fragilities, is known to help maintain the balance of immune cells. Immunoreceptors commonly recognize these molecules and other similar ones on multiple host tissues. A healthy and diverse microbiota can produce abundant metabolites, affecting the host’s signaling processes. Three significant metabolites are known to have far-reaching effects: short-chain fatty acids, secondary bile acids, and the amino acid tryptophan [73]. The fermentation of dietary fiber can also produce the propionate, butyrate, and SCFAs [74]. SCFAs can also promote various beneficial effects on the host, such as reducing the risk of diabetes and obesity. They can also help maintain the development of brain development. Some bacteria can also convert the bile acids from the liver into a substance that the host can ingest. These secondary metabolites have a wide range of effects on the host, including the endocrine signaling that affects the functioning of the liver-gut axis and metabolic homeostasis [72]. The metabolites of indole, which are found in the amino acid tryptophan, can affect various aspects of the human body, such as neurotransmitter expression, hormone secretion, and inflammation [75].

### 2.2. Food Intake and the Gut-Brain Axis

The gut-brain axis represents a duplex interaction channel that links and transmits signals between the central nervous system (CNS) and the gastrointestinal tract (GI). This interaction keeps the brain updated and informed about the components and levels of energy status in the periphery. Homeostasis is achieved after the CNS sends feedback to the gut. This complex interaction employs two metabolic routes; sympathetic and nervous pathways [76,77,78,79,80].

As shown in Figure 1, the communication between intestinal Tregs and gut-innervating neurons is carried out by the neuronal cells from the PNS, which innervates the intestine. Additionally, the enteric nervous system (ENS) in the colonic myenteric and submucosal plexus do relay physiological information to the PNS and CNS. Acetylcholine that activate colonic antigen-presenting cells, are known to assist in the development of Tregs in the gut. Vasoactive intestinal peptides (VIP) are also known to increase Tregs. Serotonin has also been shown to modulate GI and CNS functions and bone development. Thus, serotonin forms a crucial component of the gut-brain axis. Tryptophan production by gut microorganisms or the degradation of tryptophan has been shown to affect the presence of serotonin in both the CNS and the intestines. SCFAs are microbial metabolites that can promote serotonin. These metabolites can be transported through the bloodstream to the brain and the bone to regulate the nervous serotonergic system and bone tissue development. The Arc neurons are responsible for sensing the presence of metabolic cues in the periphery, which then send these molecules to the various organs in the body; such metabolic cues in the periphery include Ghrelin, PYY, and GLP-1. They interact with the brainstem, which is responsible for providing information to the CNS. These interactions play a role in the development of gut peptide receptors, such as the GLP1R and CCKIR. The various peripheral actors such as GIP, leptin, ghrelin, and insulin are involved in the regulation of food consumption, which directly affects brain development, brain function, calcium, phosphate, magnesium, vitamins (e.g., B, and K) presence in the blood and the bone (see Figure 1).

## 3. Physiology of Homeostatic Regulation of Food Consumption

The hypothalamus is the principal center of homeostatic for food consumption. The hypothalamus contains essential neurons used in modulating the energy requirements relying on both biological and environmental signals [81,82]. The arcuate nucleus of the hypothalamus masterminds homeostatic feeding using varied populations of neurons: neuropeptide Y (NPY) and Agouti-related protein (AgRP), with the synergistic expression of pro-opiomelanocortin (POMC)(orexigenic) and cocaine-amphetamine-related transcript (CART) (anorexigenic) neurons. This group of neurons communicates and inhibits the function of each other. POMC neurons release α-melanocyte-stimulating hormones (α-MSH) when triggered [83,84]. This binds to the melanocortin 4 receptor (MC4R) found in the paraventricular nucleus (PVN), which promotes decreased food consumption. On the other hand, ArRP/NPY stimulation in the Arc, activates the relinquishing of AgRP and NPY, blocking the expression of MC4R neurons in the paraventricular nucleus of the hypothalamus [85,86] (see Figure 2). PVN is largely characterized by neurons expressing the single-minded 1 transcription factor (SIM1). The third ventricle of the hypothalamus is where the Arc and the PVN are situated strategically. The third ventricle is the fenestrae brain barrier; hence it is more permeable and thus allows peripheral signals to get into the hypothalamus and bind to the receptors located in the Arc and PVN [87,88].

Certain extrinsic factors such as trauma, heat and cold stress, environmental toxicants, pathogens, diseases, etc. can cause inflammation [89], also, endogenous factors can predispose a person to inflammation, e.g., autoimmunity [90]. Additionally, the immune response to pathogens, e.g., bacteria, viruses, elicits an intracellular signaling cascades through the activation of Toll-like receptors and the higher expression of interferons, cytokines, and other immunological mediators [90,91]. Some pro-inflammatory mediators such as IL-1α, IL-1β, IL-2, IL-6, IL-8, IL-12, TNF-α, INF-γ which are crucial in starting and modulating inflammatory reactions are known to be activated by the gut-microbiota, moreso, anti-inflammatory mediator such as IL-4, IL-5, IL-10, TGF-β, has been demonstrated as inflammation biomarkers [90,92,93].

Arc neurons are responsible for circulating signaling molecules by sensing metabolic cues from the periphery; these signaling molecules include Ghrelin, PYY, leptin, and GLP-1. The Arc not only interacts with the brainstem, such as the NTS (where the vagal afferent fibers stop), which collates information from the periphery but also plays an active role in expressing gut peptide receptors such as the cholecystokinin receptor 1 (CCKIR) and GLP-1 receptor (GLP1R). GIP, leptin, insulin, ghrelin, PYY, and GLP-1 are the actual peripheral actors described in the homeostatic regulation of food consumption [94,95]. The pancreas is the factory of insulin production and secretion in response to the pre-prandial rise in blood glucose and the aid of circulation; it reaches the vantage tissues throughout the body and the brain. Though the primary activities of insulin in the periphery regulate glucose absorption by the tissues, most of the transport of glucose happens without employing glucose. Insulin tames ArRP/NPY neurons in the brain and catalyzes POMC neurons in the Arc and subsequently influencing anorexigenic effects [96]. Fat mass signals that participate in hypothalamic regulations of food consumption were proposed in 1953 and were later found with the discovery of leptin (deduced from adipocytes in quantities proportional to the fat mass) in 1994. These hormones are released into the circulatory system, which the brain communicates with [97]. Leptin plays a similar role as insulin by taming AgRP/NPY neurons and catalyzing POMC neurons in the Arc, such as anorexigenic effects. On the flip side, ghrelin is the hormone involved in orexigenic processes synthesized by the stomach and facilitates the synthesis of AgRP and NPY in the Arc, which induces its orexigenic effects. Though ghrelin also inhibits POMC neurons, its receptors are not known to be expressed in POMC neurons [98]. It was hypothesized that ghrelin inhibitory prowess may be attributed to the activation of NPY/AgRP neurons communicating and taming POMC neurons [99].

Other areas of the intestines also synthesize peptides aiming at the hypothalamus to regulate the consumption of food. EECs form pathways with the intestines; to the colon from the duodenum; all participate in the synthesis and secretion of GI peptides, with some participating in the homeostatic modulation of food consumption. CCK cells are EECs arranged along the intestines, most found in the duodenum. In the intestinal lumen, CCK is produced in response to nutrients mainly proteins and lipids which influence anorexigenic effect through the POMC neuron activation evinced in the cells of the nucleus of the solitary tract, via the vagus nerve [100,101,102,103].

The jejunum and duodenum are the sites of glucose-independent insulinotropic peptide (GIP) synthesis, primarily released in response to ingesting nutrients such as glucose or fats and its influence on food consumption. This phenomenon is mediated by GLP-1 (some other anorexigenic peptides produced by EECs) found at high densities in the ileum and distal colon. The production and secretion of GLP-1 are influenced in response to the availability of nutrients by direct or indirect effects regulated by the GLP [104,105].

## 4. Non-Homeostatic Regulation of Food Consumption

Individuals’ quest to consume food for pleasure is a critical aspect; non-homeostatic signals play a critical role in stimulating this phenomenon. The reward system of wanting food or drugs is also influenced by dopamine in the mesocorticolimbic pathway [106,107]. This influence can lead to overconsumption of food and may subsequently lead to obesity. Curtailing the dopaminergic pathway will slow or stop the drive for food during an operant task. The synthesis of dopamine is in two stages; tyrosine is first produced by a rate-limiting enzyme (the tyrosine hydroxylase (TH)), which results in the production of dihydroxyphenylalanine (DOPA). The second stage involves the production of aromatic L-amino acid decarboxylase dopamine. These products are stored in vesicles before being released into the synaptic space. Dopamine influenced reward system leads to three physiological demands of food consumption such as learning, liking (controlled by GABA and the opioid system), and wanting (controlled by the dopaminergic system) [108,109,110].

## 5. Sympathetic Routes

The immune system is built in the gut- one of the largest divisions that house various kinds of cells. Furthermore, the gut represents the largest endocrine system; the intestinal EEC represents only 1% of the cells. The intestines carry a rich reservoir of peptides, and about thirty different hormones have been attributed to the GI acting at lengths from the gut and other organs [111,112]. EEC is found nearly everywhere in the GI tract and is a part of the essential components of the endocrine organs that regulates food consumption [113]. EECs release three major metabolites in pre-prandial states: CCK, PYY, and GLP-1, which all act as anorexigenic hormones. These hormones mainly target the brain after they have been released in the blood or when they trigger the vagal afferents nerves [114,115].

## 6. Nervous Routes

The GI tract is the brain of the gut (second brain) and provides the largest surface for intestinal neurons after those of the CNS [7,116]. The major player of the GI tract is the afferent fibers, responsible for conveying sensory signals from the top of the GI tract to the principal CNS using vagal and splanchnic routes. These two routes (splanchnic and vagal) are essential satiation modulators. The link keeps the brain abreast regarding energy levels from the periphery. The vagal afferent represents the most delineated afferents in the regulation of food consumption. The vagal afferent originates from the intestines with their cell inclusions in the nodose ganglia and projects to the nucleus tractus solitarus (NTS) in the brainstem [117,118]. In addition to the hypothalamus, the NTS is made up of projections to several regions of the brain. Vagal afferents act as mobility or distension satellites and evince a vast array of GI hormone receptors such as GLP-1, CCK, PYY, or ghrelins [119,120]. The activation of distinct populations of the vagal afferents is influenced by specific stimuli from the GI tract. The major role of the vagus nerves, as was demonstrated by a group of researchers, is its modulation of non-homeostatic food intake in the gut-brain axis. The researchers demonstrated this with the aid of optogenetic functional pathways that links the gut to the brain reward system, which includes the vagus nerve, the nodose ganglia, NTS, and the dorsal striatum [121,122,123].

### The Gut Microbiota a Major Player in Nervous Systems

The gut microbiota plays a critical role in nervous system function. It is believed that its presence can regulate the ENS development in both adult and postnatal animals. The ENS controls the signals and intestinal movement to the CNS. In certain animals, the myenteric plexus of germ-free mice is abnormal. This situation is due to decreased nerve density and the number of cell bodies per ganglion. In addition, gut microbiota can affect the development of enteric glial cells. These cells regulate ENS by connecting the gut and the brain. In 2015, a study revealed that the presence of gut microbiota could regulate the flow of homeostatic and colonization factors in the intestinal mucosa of mice [124,125,126]. The number and density of enteric glial cells in germ-free mice are significantly decreased compared to that of normal animals. This suggests that the presence of microbes and microbiota in the gut can affect the development of the gastrointestinal tract. Furthermore, enteric cells play a role in the regulation of the nervous system by linking microbial cues to the host’s nervous system [126]. Although it is known that the presence of enteric cells can affect the development of the nervous system, further studies are needed to determine the exact relationship between these cells and disorders such as neuropsychiatric disorders [127,128].

It has been shown that restoring the balance of the gut microbiota can help manage stress and anxiety [129,130]. For instance, serotonin-associated behavior modification with the treatment of *L. rhamnosus* can reduce anxiety and depression by decreasing the activity of the vagus-dependent pathway and this effect is linked to the increased expression of GABA in the hippocampus [131]. In another study, they revealed that treating the gnobiotic mice with the bacteria known as *Blautia coccoides* reduced their anxiety levels. On the other hand, the treatment with *B. infantis* did not affect the anxiety levels [132]. In another study, the researchers noted that a probiotic treatment can help improve the behavior of *C. rodentium* and *T. muris* [133,134]. Studies have implicated the effects of probiotic supplements such *as Bifidobacterium animalis subsp. lactis*, *Streptococcus thermophiles, Lactobacillus bulgaricus,* and *Lactococcus lactis subsp. lactis* on the functioning of the midbrain of women [135]. Interestingly, literature have demonstrated that the administration of *L. rhamnosus + L. helveticus* to *C. rodentium* before and after infection blocks memory dysfunction in the mice [136], therefore buttressing the importance of the microbiota-gut-brain axis relationship.

In neurodegenerative diseases, such as Parkinson’s and Huntington’s disease, the survival of neurons can be regulated by certain types of microRNAs (such as Mir433 and Mir9, Mir145) and these MicroRNAs are can be modulated by gut microbiota [137,138,139]. The presence of microbial products in the environment can affect the functioning of the nervous system. For instance, the butyrate can induce the production of Mir375 in human embryonic stem cells [140]. It is believed that this substance can act as a histone deacetylase inhibitor. Histone deacetylase 3 is required for the development of intestinal epithelial cells and the maintenance of a balanced microbiota [141]. A study on a high-fat diet (HFD) revealed that the presence of saturated free fatty acid could influence the function of enteric neurons. In a mouse model, the apoptotic and palmitate-induced contraction of enteric neurons was mediated by the up-regulation of Mir375. It has been demonstrated that the presence of a gut microbiota that is unhealthy is associated with neuropsychiatric disorders [142,143]. It is also possible that there exists a link between the gut microbiota and the regulation of neural function.

## 7. Enteric Nervous System and the Microbiota’s Molecular Patterns and Signatures

The Pattern Recognition Receptor (PRRs) and the Microbe-Associated Molecular Pattern (MAMP) are believed to be involved in regulating microbiota in the human gut. Crosstalk between PRRs and the MAMP of the microbiota is essential to the host’s recognition of the bacteria. It is interesting to note that some of the intestinal bacteria, such as *Lactobacillus* [131], *Bifidobacterium* [144] *and Blautia* [132] which are commonly used as probiotics, are Gram-positive. It has been known that stress exposure can reduce the abundance of certain types of bacteria in the human gut. Some of these are known to be Gram-positive, such as the bacteria that are enriched after an antibiotic treatment. These are known to produce a type of acid that is associated with the development of the receptor TLR2. Studies have shown that the presence of the TLR2 protein in the gastrointestinal tract is linked to the function of the epithelium and the ENS. In C57BL/6J mice, the lack of this protein significantly affects the neurochemical profile and architecture of the ENS. In addition, the decreased levels of GDNF in the smooth muscle cells are associated with the development of intestinal dysmotility. The presence of the DTLR2-GDNF axis suggests that the integrity of the ENS is dependent on the presence of the microbiota [145]. In addition, ENS dysfunction can increase patients’ sensitivity to chemical-induced colitis. It is believed that the presence of dysbiosis in the microbiota can contribute to the development of Crohn’s disease [146]. The relationship between the lipotoxin and receptor 2 (TLR2) is believed to be involved in regulating the gut-brain axis. A recent study revealed that the treatment of schizophrenia with a high level of lipotoxic acid significantly decreased the expression of two phosphoproteins, which are known to be up-regulated in the development of schizophrenia [147].

## 8. Significance of Gut Bone Axis and Bone Metabolism

The gut microbiome is a complex system that regulates the homeostasis of a person’s body. Its interactions with other organs and the environment can affect the development and maintenance of bone health. The gut microbiota-bone axis is a relationship between the various metabolites that are produced by the microbes and the skeletal homeostasis of a person [148]. The emerging understanding of the importance of microbes in the development of bone mineral density suggests that they can alter the function of the gut’s signal pathways. Some key endocrine regulators involved in this process include the parathyroid hormone and the calcium-sensing receptors (CaSRs). The CaSRs are located in different human body tissues, including the kidney, intestine, and stomach. Their importance is acknowledged as the body’s control over calcium homeostasis is dependent on their dynamic interactions with other organs [149,150,151,152,153]. As the gut is a major source of nutrients, such as vitamins B and K, the organs can directly or indirectly affect the development and maintenance of bone mineral density. The non-collagenous protein known as osteocalcin is abundant in the bone matrix. Vitamin K is a vital component of this protein, which is also known as bone Gla-protein [153,154,155]. It is required for the activation of this process. The carboxylation of this component is very important for binding this protein to bone minerals. Although vitamin K can be obtained from dietary sources, the production of vitamin K through the intestinal tract is an essential component of the vitamin supply. If the gut microbiota cannot maintain its balance of beneficial microbes, the reduction in vitamin K production could lead to an oversupply of uncarboxylated osteocalcin in circulation. When carboxylated osteocalcin is not present in the bone matrix, it can weaken the bone tissue and make it vulnerable to fracture [155,156].

The presence of butyrate in the gut microbiota can influence the actions of T regulatory cells, which can lead to the formation of bone and the proliferation of osteoblasts (38). These effects can also prevent bone density and bone loss [157,158]. According to a study conducted by Stefano et al., the growth of gut bacteria can affect mineral bone density [159]. It was found that bone loss at the lumbar spine and the femur neck was associated with the development of intestinal bacterial overgrowth. This could be a predisposing factor for osteoporosis. In addition to malabsorption, this overgrowth could also lead to the metabolism of nutrients, such as vitamin K and carbohydrates, which could affect the control of osteogenic events [160].

## 9. Promising Health Benefits of Probiotics on Bone

Various studies have shown that probiotic administration can help prevent bone loss and preserve bone mineral density. Factors that have been identified as beneficial include the combination of probiotic strains, such as those from *L. reuteri, L. rhamnosus GG, L.paracasei and L. plantarum as well as Bifidobacterium longum*, a prebiotic [161,162]. In a study conducted on mice, researchers discovered that probiotic (*L. reuteri*) treatment significantly reduced the incidence of bone loss [163]. They also noted that the effects of the probiotic on the bone resorption mechanisms were observed to be very beneficial. Some of the key factors that were identified as being beneficial include the reduction of the TRAP 5 and RANKL levels. The results of the study support the idea that the probiotics can help prevent bone loss. It also noted that the reduction of osteoclastogenesis was observed in the animals that were treated with *L. reuteri*. In addition, the number of helper T cells was also increased in the animals that were ovariectomized. The study also noted that the effects of the probiotic on bone metabolism were observed to be very beneficial. They noted that it inhibited the activation of the T-cell signals related to bone formation [163].

In a different study conducted on male mice, the authors noted that the oral probiotic, *L. reuteri*, significantly improved their vertebral bone mass, femoral bone mass, and overall body composition [164]. They also found that it reduced intestinal inflammation. The researchers analyzed the RNA profiles of the samples and found that the probiotic inhibited the production of basal tumor necrosis factor (TNF) mRNA. The administration of *Bifidobacterium longum* increased serum osteocalcin levels and favored bone formation parameters. In addition, Parvaneh et al. found that the reduction in the telopeptide levels and the bone resorption parameters were related to the reduction in the microstructure of the femur. The effects of the supplement on the bone mineral density of ovariectomized rats were also shown. The development of bone health is a crucial factor for human health. It involves the upregulation of the two key genes known as the BMP-2 and the SPARC. These supplements have been shown to upregulate these crucial genes [165]. Recent studies revealed that the probiotic, *L. reuteri*, can stimulate the production of various immune cells that promote bone health. These cells were able to decrease the levels of osteoclastogenic cytokines (Interleukins 6 and 17, TNF-α) and anti-osteoic cytokines (Interleukins 4 and 10, IFN-γ). In the same study, *Lactobacillus rhamnosus* was shown to suppress osteoclastogenesis and regulate Treg-Th17 differentiation and distort Treg-Th17 cell balance, thereby influencing bone metabolism [166]. The researchers noted that the presence of the probiotic could suppress the development of osteoclastogenesis and increase the level of the protein that is involved in the regulation of bone metabolism. In addition, it can affect the development of bone resorption capacity by modulating the RANKL and BMP pathways. These findings suggest that the use of probiotics can improve the function of the bone resorption system [167].

According to studies, the beneficial effects of probiotic supplements are mainly due to the stimulation of the production of butyrate by the microbes. VSL#3 (comprising of *Bifidobacterium longum*, *Bifidobacterium infantis*, *Bifidobacterium breve*, *Lactobacillus plantarum*, *Lactobacillus acidophilus*, *Lactobacillus bulgaricus*, *Lactobacillus paracasei*, and *Streptococcus thermophilesis*) is a group of bacteria that can help boost the production of butyrate [168,169]. These studies showed that the use of probiotic supplements could also help prevent bone loss by improving intestinal permeability and reducing inflammation. It can also help maintain the health of the bone marrow and intestines.

In a study conducted by Muccioli et al., they discovered that the application of prebiotics to the gut significantly improved the function and structure of the intestinal barrier [170]. They also found that the increase in GLP-2 levels triggered the synthesis of proteins that are crucial for the development of tight junctions. Prebiotics are known to have various effects on the development of the intestinal barrier. One of the most important mechanisms that are involved in this process is the concentration of SCFAs in the intestine. This increase in SCFA levels can help promote the growth of beneficial bacteria and prevent the development of pathobionts [171]. In addition to improving the function of the intestinal barrier, SCFAs can also stimulate the production of immunosuppressive cytokines and increase the levels of intestinal mucus. They can also help reduce the levels of proinflammatory mediators. Literature shows that the administration of butyrate, propionate and acetate as a mixture or as acetate alone can increase the number of T cells and the level of IL-10 in the interstitium. According to Smith and colleagues, SCFAs can exert their effects by inhibiting histone deacetylases. They also found that these compounds can trigger a process that is related to the formation of GPR43 [172].

## 10. Underlying Mechanisms of Bone Metabolism by Gut Microbiota

### 10.1. Gut Microbiome and Its Potential Role in the Mineral Uptake of the Bone

The metabolism of the bone affects the availability and presence of calcium. One of the most common forms of calcium is calcium hydroxyapatite, which is found in the bones and teeth. The body must maintain a healthy balance of this mineral, as it can contribute to various metabolic functions. Calcium and vitamin D are transported through the walls of the small intestine’s paracellular and transcellular mechanisms. The microbiota in the intestine helps in the absorption of these two essential nutrients. Prebiotics are nutritional supplements that can help improve the health of the gut microbiota [173,174]. The beneficial effects of prebiotics can also be seen in the release of short-chain fatty acids (SCFAs) into circulation. The interactions between the SCFAs and proteins are known to contribute to the regulation of the metabolism of the bone. They also help in the absorption of calcium. Due to the reduction of the intestinal pH, the SCFAs interact with the proteins to increase the levels of calbindin-D9k and TRPV6 [175,176]. Despite the various mechanisms that are involved in the development of osteoporosis, the gut microbiome can still play a role in preventing the development of this condition. The small intestine’s transcellular active transport system is responsible for absorbing calcium. The ability of vitamin D to facilitate passive diffusion via the paracellular gradient over the intestine was also known to contribute to the absorption of calcium. This finding suggests microbes in the gut play a role in the prevention of osteoporosis [177]. Prebiotic fibers could be an effective alternative to the calcium absorption pathway as this can help maintain a favorable microenvironment for the bone. In a separate study, the gut microbiome’s effects on bone development and maintenance were shown. In the models employing Xenograft models, the changes in the gut microbiome during the early stages of life were investigated, and they showed that the gut microbiome could compromise the strength of the bones [178]. As shown in Table 2, the major gut microbes and their protective effects on bone and brain has been depicted.

### 10.2. Crucial Role of Gut Microbiome in Preventing Bone Loss—Bone Metabolism and Immune System

Studies have shown that the gut microbiome can regulate the activities of osteoblast and osteoclast cells [194]. The complex network of the skeletal system and immune system helps the body fight off invading pathogens [195]. T cells, which are known to play critical role in bone health, are also involved in this process [196]. The diverse immune cells that are involved in this process can regulate the activities of osteoblast and osteoclast cells. They can also translocate bone tissue. When a subset of T lymphocytes is activated at the intestinal lining, it moves to the bone marrow, where it influences the bone remodeling process [197]. The RANK-RANK ligand interactions and the downstream signaling that these processes play can help in the development of bone remodeling mechanisms [198,199]. Some of the most common factors that trigger the development of osteoclastogenesis are the cytokines, such as tumor necrosis factor and Interleukin 1, 6, 8, 11, 15, and 32. However, other factors such as Interferons α, β, and γ and Interleukins 10, 13, 18, 33 can also hinder the activity of osteoclast cells [200]. The lack of estrogen is known to cause bone loss. In studies, it has been shown that the gut microbiome can regulate the responses of the immune system to inflammatory threats. In order to understand the effects of sex steroid deficiency on the development of bone loss, studies were conducted on mice that were free of germs or bacteria. These studies revealed that the animals were more prone to experiencing trabecular bone loss. The link between gut microbiota and estrogen have a direct effect on the development of skeletal integrity and gut health. In a study, researchers discovered that the treatment of the bacteria, known as *Lactic acidophilus rhamnosus*, reduced the inflammation of the intestine and bone marrow and decreased the risk of bone loss. The results of the studies revealed that the treatment of the bacteria, known as probiotics, reduced the inflammatory pathways and gut permeability in the mice. This suggests that the beneficial effects of the gut microbiome can be carried out in the prevention of postmenopausal osteoporosis [201].

### 10.3. Effects of Calcium-Regulating Hormones

The role of calcium-regulating hormones in the development of healthy bone is known to be important. There are various hormones that regulate bone metabolism [202]. It is also believed that the gut microbiome can provide a net anabolic effect to the skeleton by supplying it with insulin-like growth factor (IGF-1). A pattern of longitudinal growth characterizes the development of juvenile bone in mammals. This process depends on the presence of IGF-1 and the paracrine and endocrine mechanisms. The interactions between the host and the microbiota can influence the production and metabolism of this growth factor [203]. The role of the parathyroid hormone in the development of skeletal muscle is known to be important. It is believed that the presence of a metabolite produced by the microbiota can help stimulate the formation of bone [204,205] (See Figure 3).

Serotonin, also known as 5-HT, is a neurotransmitter that affects skeletal metabolism. It can be triggered by its receptor gene, which is known as the serotonin transporter. Phytoestrogens are compounds that have similar structural properties to endogenous estrogens. They can act as natural estrogen receptor antagonists [206]. Daidzein is a well-known phytoestrogen that is active in converting into a metabolite called equol in the body. It has been shown that this bacterial compound can improve the estrogenic effect of bones [207] (see Figure 4).

### 10.4. Postmenopausal Osteoporosis and Gut Microbiome-Targeted Treatment Strategies

Maintaining a healthy and adequate estrogen level is very important to maintaining the diversity of the intestinal microbial community. The gut microbiome is composed of various types of bacteria, including *firmicutes, proteobacteria,* and *actinobacteria* [208]. The *firmicutes/bacteroidetes* ratio of these organisms is associated with the homeostasis of the gut microbiome [209]. A reduction in the gut microbiota’s diversity and an increase in its *firmicutes/bacteroidetes* ratio are known to cause dysbiosis. This condition can trigger inflammation. It is also linked to the development of osteoporosis [210]. In a typical gut, there is a balance of the intestinal microbial community, thus preventing the development of harmful microorganisms. During a disease, the pathogens in the gut can penetrate the intestinal barrier and trigger an immunological response. This can then lead to bone resorption by osteoclasts. The reduction in the gut microbiota’s diversity can also lead to the development of osteoporosis. This condition is characterized by the loss of bone mass and the fragility of the skeleton. One solution for osteoporosis is to introduce probiotic supplements to help restore the balance between the gut microbiota and the host [211] (See Figure 3 and Figure 4).

### 10.5. Estrogen as Critical Factor of the Gut’s Microenvironment

Estrogen is a vital component of the microenvironment of the gut, which regulates the development and maintenance of the gut’s microenvironment. It can exert its effects through increased levels of glycogen, mucus secretion, and epithelial thickness. Many phytoestrogens are known to exert pro and anti-estrogenic effects on the target tissue [212]. Due to their ability to promote the development and maintenance of the gut microenvironment, phytoestrogens are commonly used as therapeutic agents for hyperestrogenic diseases [213]. Intestinal bacteria can also produce equol, which can affect the reproductive tract. Equol usually is present as a diastereoisomer, and intestinal bacteria are specific in synthesizing the S-(-)equol enantiomer exclusively. This enantiomer shows a selective affinity for estrogen receptor-β and thus has been vigorously pursued in clinical and pharmacological research as a nutraceutical and pharmacological agent [214]. In an in vivo study, male mice that were fed with equol showed lower uterine weight and thinner vaginal epithelial tissues. The various tissues of the brain, intestine, and abdomen express estrogens, which can influence the development of neural responses [215]. Estrobolome is a gene in the gut’s microbiome that plays a role in the metabolism of estrogens and phytoestrogens [216]. Beta-glucuronidase from microbes can break down estrogens and phytoestrogens through its actions on the estrogen receptor alpha and beta. The interactions between estrogens and estrogen receptors can trigger downstream signaling pathways that can affect the development and maintenance of the target tissues [217]. High-speed sequencing techniques have highlighted the importance of the gut microbiome in developing and maintaining skeletal metabolism. Through a combination of host factors, such as immunomodulation, hormonal control, and mineral absorption, the metabolism of bone can be regulated [218]. The composition of the gut microbiota is influenced by various factors, such as diet and antibiotics [219]. In order to control the development and maintenance of bone mass, the gut microbiome can be influenced by a variety of mechanisms. Some of these include the growth of beneficial bacteria, as well as the modulation of estrogen and bone mass through the use of prebiotics [220,221].

## 11. Gut Bone Axis and Cannabinoid

The highly regulated bone remodeling events require numerous hormones and messenger molecules. The principal mechanisms behind the systemic and the local level of molecular events remain vague. The function of the elements of the endocannabinoids system (ECS), such as endogenous cannabinoid ligands, cannabinoid receptors (CBRs), and the enzymes accountable for endogenous ligand synthesis and degradation, is exceptionally fascinating. The link between the ECS and bone health speculates the possible application of cannabinoid receptor ligands for the treatment of bone diseases related to enhanced osteoclastic bone resorption, like osteoporosis and bone metastasis [222]. Studies revealed that the endocannabinoid system is identified to exhibit diverse effects in controlling multiple physiological events, such as immune responses, pain sensitivity, appetite regulation, and energy maintenance. In addition to these roles, the endocannabinoid system has also been involved in regulating bone metabolism. Hydrolysis of membrane phospholipids generates cannabinoids endogenously, which are hydrophobic in nature. These molecules can combine with both plant-derived and synthetic cannabinoids. Also function together with receptors such as cannabinoid receptors type 1 (CB(1)) and 2 (CB(2)) and the GPR55 receptor to modulate cellular activities via a range of signaling pathways. Studies conducted in mice with targeted inactivation of cannabinoid receptors exhibited that cannabinoids show a vital role in the modulation of bone metabolism CB(1) deficiency mice have high peak bone mass due to osteoclast defect. However, they develop age-related osteoporosis owing to the diminished bone formation and formation of bone marrow fat. In contrast, CB(2) deficient mice have shown comparatively normal peaks of bone mass. Nevertheless, they develop age-related osteoporosis owing to the augmented bone turnover by uncoupling bone resorption from the bone formation. Similarly, GPR55 deficient mice have high bone mass as a result of defective osteoclast resorption, although bone formation is found unaffected [223].

Individuals with spinal cord injury (SCI) endure an acute loss of bone minerals below the level of the lesion. Studies conducted in a rat model of spinal cord injury to investigate the therapeutic effect of cannabidiol on sublesional bone loss showed that CBD treatment accelerated the serum levels of osteocalcin, decreased the serum levels of collagen type I cross-linked C-telopeptide, and augmented bone mineral density of tibiae and femurs. SCI rats treated with CBD revealed higher bone volume, trabecular number, trabecular thickness, lower trabecular separation in proximal tibiae, elevated ultimate compressive load, and energy to the max force of femoral diaphysis. Treatment of SCI rats with CBD exhibited upregulated mRNA expression of osteoprotegerin, alkaline phosphatase, wnt3a, Lrp5, and CTNNB1 and downregulated mRNA expression of receptor activator of NF-κB ligand (RANKL) and tartrate-resistant acid phosphatase (TRAP) in femurs [224].

## 12. Cannabidiol (CBD)

CBD is considered a non-psychotropic component of the *Cannabis sativa* (hemp) plant that, in the recent decade, has hugely attracted the attention of many people in health and pharmacy [225,226,227]. Currently, the only successfully approved CBD drug by the US Food and Drug Administration is EPIDIOLEX^®^. This drug is used for the treatment of drug-resistant pediatric epileptic seizures caused by several rare syndromes. However, they are no single dietary supplement or its ingredients approved by the US Food and Drug Administration [228,229,230]. The marketing of CBD products is hugely growing and is worrisome partly due to a lack of clear federal regulations and quality inadvertence. This menace has threatened and may lead to adverse health impacts on the general public. Therefore, several questions that seek urgent answers have been stringently proposed [231,232]. Some of these are; what medical role does CBD play? Can it serve the role of a non-prescriptive drug? What doses seem adequate? What influence has it had on the gut microbiota? Is there any difference between CBD extracted from hemp and isolated CBD? Who is at risk of CBD administration? Is there any future for hemp products, and what is the direction [1]?

Given these, we seek to find answers to the above reasonable questions by first putting together a review that highlights some literature that already talks about CBD and the gut microbiome. The entire write-up of this review seeks to talk about the CBD modulation of the microbiota, gut-brain, and gut-bone axis.

A study by Talamantes et al. hypothesized that due to the increased interest in cannabidiol and its derivatives, cannabidiol is likely to be accumulated in the environment, which could have a dramatic effect on aquatic and environmental microorganisms and destabilize their microbiome. They used zebrafish larvae as a model organism. They found that cannabidiol oil, when exposed passively to the larvae at concentrations as high as 200µg/L, did not influence the survival of the larvae and had minimal effects on the host-associated microbiome. However, they observed a minimal disturbance in the sequence of *Methylobacterium-methylorubrum spp., Chryseobacterium sp*., and *Staphylococcus spp*. [9].

CBD has gained massive rising attention because it contains bioactive components that are believed to have a potential modulatory effect on the gastrointestinal tract [233,234,235]. Research on the therapeutic benefits of CBD has subsequently led to many discoveries of novel drugs with health benefits. Some of the health benefits of CBD outlined so far are; its ability to modulate food consumption, gastric secretion, gastro-protection, nausea, emesis, iron transport, intestinal inflammation, gastrointestinal tract mobility, and cell growth in the gut [236,237,238]. The role of CBD in anti-inflammatory activities has been extensively explored in human medicine [239]. Nonetheless, CBD’s modulatory role on the gastrointestinal tract activities, such as immune competency, especially in veterinary medicine, is not defined due to the challenges in prescribing doses of CBD for the treatment of animals.

Konieczka et al. experimented on the multifocal mechanisms influencing the cross-talk of *Clostridium perfringens* and its host response (which is an imperative need in poultry husbandry). They investigated the profile of CBD regulation of the host response to *Clostridium perfringens* by infecting chickens with the bacteria. They saw that the infected chickens did not exhibit any clinical manifestations indicating the possible hazards of the transmission of the pathogen to the food chain in commercial sectors. Also, they realized that CBD affected the chickens’ response to *Clostridium perfringes*, indicating the positive activity of CBD in the upregulated action of the genes that ascertain gut barrier role. CBD boosts the shifts in the enzyme activities of gut bacteria [10].

Abi et al. researched CBD and Omega 3 potential remedies for high-fat alteration of gut microbiota impacts on learning, memory, and anxiety response in mice. They employed 15 mice put into three groups-group one; which administered water and ordinary chow *ad libidium*, group two; which administered two high-fat diets (HFD) and water *ad libidium*, group three; which administered three had HFD 10 mg/kg CBD and 300 mg/kg omega 3. These groups were allowed for twelve weeks before they were tested on an elevated plus maze, Y-maze, and spontaneous alteration measurements. *E Coli* count from the mice’s fecal matter was performed after slaughtering. They observed that CBD and omega 3 group was significantly longer than the control group at a *p* < 0.05. Their findings, in conclusion, stipulated that HFD-enhanced gut E. coli overgrowth was reduced by CBD and omega 3 and that CBD and omega 3 also curtailed the memory impairment and anxiety induction by HFD [5].

Alteration of the gut microbiota can cause inflammatory bowel disorders and behavioral changes. A group of researchers worked on fish oil, CBD, and gut microbiota using a murine model of colitis. They found in the study that the administration of CBD or fish oil decreases inflammation. They also investigated the synergistic activities of CDB/Fish oil on inflammation and dysbiosis in the dextran sulfate sodium (DSS) model of mouse colitis (which causes behavioral imbalances). They realized that administering CBD alone did not produce any effect at any of the dose levels tested, though fish oil administration at concentration levels of 35 mg/mouse, 50 mg/mouse, and 75 mg/mouse produces reduced inflammation at day 8. Nevertheless, the combined administration of CBD (3 mg/kg, 10 mg/kg) and fish oil (75 mg/mouse) was able to attenuate the inflammation. All the inflammatory markers were attenuated and also increased the intestinal permeability at both day 8 and day 14 at the combined concentrations of CBD (1 mg/kg) and fish oil (20 mg/mouse), but no significant attenuation was observed when administered individually. They concluded in their study that Fish oil; CBD, and their combination influenced the gut microbiota. Specifically, *Akkermansia muciniphila* was increased by Dextran Sulfate Sodium (DSS) on day 14, but CBD and fish oil combined therapy was drastically elevated at all treatment levels of day 8 [3].

Another study focused on the therapeutic benefits of CBD on Parkinson’s disease (PD) using a genetically mutated mouse model, analyzing gut-brain metabolism. They aimed to identify the effects of CBD on PD by exposing the PD genetically manipulated mouse model to CBD and subsequently estimating both motorial and postural coordination through a modified swim test. The researchers used histopathology of substantia nigra and the gut-brain metabolic analysis via UHPLC-TOF-MS approaches. It was pleasing to observe that CBD significantly improved motor deficits of the PD model and protected the substantia nigra area. Highlighted in the pathological and therapeutic process in line with the gut-brain axis was the metabolic function of fatty acid biosynthesis, butanoate metabolism, and pantothenate/CoA biosynthesis. They concluded remarkably that CBD could attenuate PD through the neuroprotective effects on the midbrain [8].

It is now clear these compounds also benefit patients with neuroinflammation [240,241]. A study that investigated the role of gut microbiota in treating inflammation and paralysis in a mouse model demonstrated that treatment with CBD and THC significantly decreased inflammation levels. They also exhibited a significant increase in the number of anti-inflammatory cytokines. The presence of certain cannabinoids can suppress neuroinflammation and prevent microbial dysbiosis [241].

Using 16S rRNA sequencing, the researchers revealed that their experimental autoimmune encephalocephylitis (EAE) mice had high levels of mucin-degrading bacteria, such as the *Akkermansia muciniphila*. However, after treatment with CBD and THC, the number of these bacteria decreased significantly. Fecal material transfer (FMT) experiments also revealed that the effects of CBD and THC on the microbiome were related to EAE [241].

## 13. Conclusions and Future Perspectives

The gut microbiome is a collection of diverse microorganisms that live in the gut. Its interactions with hosts, the active metabolism, and the proximal and distal organs play a vital role in human survival. Dysbiosis is a condition that can disrupt the gut microbiota, which is believed to be the cause of many diseases, such as osteoporosis and nervous system-related conditions. Recent studies have shown that the administration of prebiotics or probiotics can restore the balance of the gut microbiome and improve bone and brain health. While more research is needed to understand the complex relationship between the gut microbiome and bone health, as well as the gut microbes and brain health, we must acknowledge the immensity of research that has already been carried out on the impact of gut microbiota on both axes (gut-brain and gut-bone axes). Due to the complexity of most diseases, such as osteoporosis and neuropsychiatric disorders, effective treatment strategies must be developed through a multidisciplinary approach. The gut microbiota has been known to regulate the behavior and function of the brain. In animal models, disruptions in the functioning of certain microbiota members were linked to neurophysiological disorders. These findings support the idea that microbiota plays a role in maintaining the brain’s health. Even though controversies exist, such as the discoveries on some of the roles of the gut microbiota in the development and maintenance of CNS-associated diseases, which have raised questions about its regulation, nonetheless, there are other possibilities of their epigenetic regulations affecting the function of neurons. For these reasons, the importance of studying the interactions between the gut microbiota and the brain has been acknowledged as a key to developing effective neurodevelopmental therapies.

Furthermore, prebiotics, synbiotics, or probiotic combinations can help maintain the balance of the gut microbiome and improve the health of the brain and the bone. This strategy is an emerging approach to managing metabolic bone or brain diseases. Further studies are needed to investigate the effects of cross-talk on the molecular targets critical for bone and brain-related diseases for therapy. In treating infectious diseases, antibiotics is key in current times; however, treatment with antibiotics leads to the depletion of unique gut microbiota, thus subsequently bringing about alterations in gut microbial composition. Additionally, prenatal exposure to antibiotics sometimes results in congenital disabilities, obesity, and unwanted metabolic consequences in childhood. Administration of antibiotics has also been demonstrated to interfere with the gut microbiota–immune interaction, thus resulting in immune interference. In a study using laboratory diet-induced obese (DIO) mice, the chronic administration of THC prevented weight gain. In the same study, it was demonstrated that DIO-mediated modifications in gut microbiota were prevented in the chronically THC-treated mice [242]. CBD on the other hand has been demonstrated to normalizes intestinal motility when the gut is perturbed by a pro-inflammatory stimulus [243,244], and also has been revealed to decrease acetylcholine- and prostaglandin F2α-induced contractions in the small intestine [243]. Purified CBD has also been revealed to attenuates inflammation at low doses [3]. In conclusion, having shown that THC has anti-obesity properties and appears to prevent modifications in gut microbiota additionally. In contrast, CBD supports gut health, normalizes intestinal motility, and shows no adverse effect on gut microbiota. Therefore, it is safe to infer that THC and CBD from Cannabis sativa have a beneficial impact on the gut microbiome, unlike most well-known antibiotics, which show detrimental effects.

More in-depth research is needed on the possible effects of THC and CBD on gut microbiota and how such translate into the modulatory capacity of gut microbiota on the gut-brain and the gut-bone axes.

## Figures and Tables

**Figure 1 metabolites-12-01247-f001:**
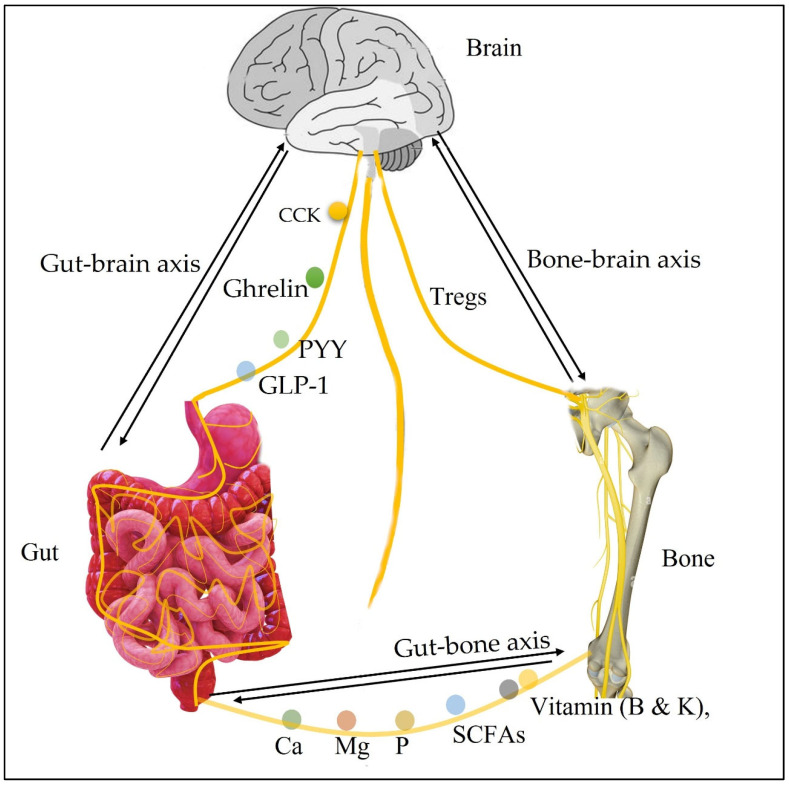
Simplified pathways in the gut-brain and gut-bone axes.

**Figure 2 metabolites-12-01247-f002:**
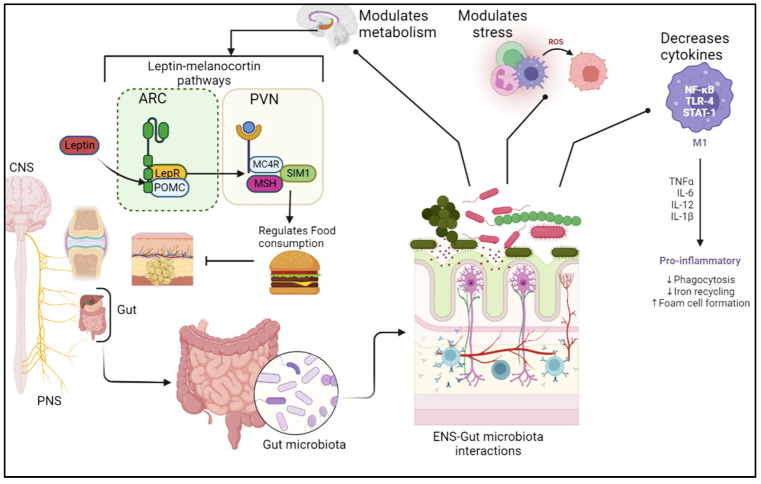
Diagrammatic illustration of leptin–melanocortin signaling pathway. The leptin-melanocortin pathway is crucial in energy homeostasis by acting as a link by which information about energy status can be relayed to the CNS to control food intake and energy expenditure. LepR, leptin receptor. SIM1, single-minded 1. ARC, arcuate nucleus of the hypothalamus. POMC, proopiomelanocortin. MC4R, melanocortin-4 receptor. PVN, paraventricular nucleus of the hypothalamus. MSH, melanocyte-stimulating hormone. POMC neurons in the ARC when activated by leptin produces α-MSH, which in turn activates MC4R which produces a satiety signal. The gut microbiota is also known to modulate stress, cytokines, proinflammatory activities and the CNS metabolism.

**Figure 3 metabolites-12-01247-f003:**
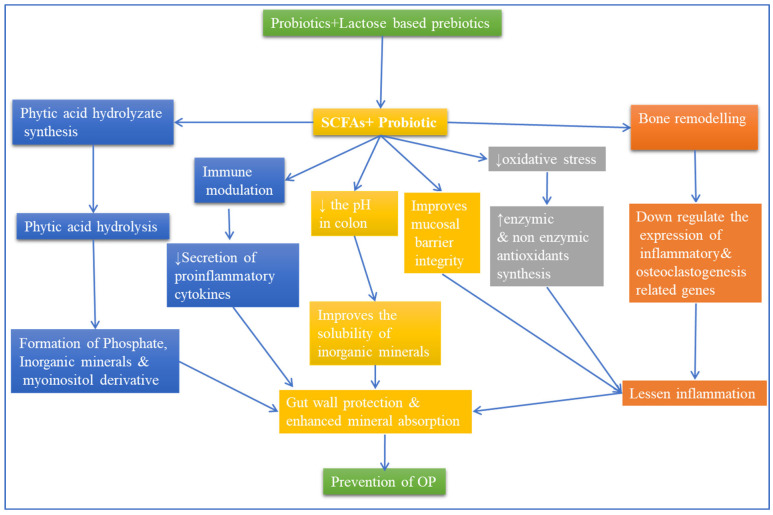
Effect of microbiota on osteoporosis and the crucial regulatory factors in bone metabolism. Beneficial microbes of the gut are influenced by diet, antibiotics, and probiotics. This impact has downstream ramifications, especially on bone mass, via multiple mechanisms. For instance, microbiota abundance does impact estrogen bioavailability which is crucial in bone mass. Secondly, microbiota does support the immune system by modulating the expression of inflammatory cytokines. Thirdly, gut microbiota can generate metabolites such as short-chain fatty acids. It has also been shown that gut microbiota does alter intestinal permeability and enhances vitamin D availability and bone mineral absorption. Finally, the gut microbiome is known to affect the gut-brain axis via the modulation of the abundance of hormones: 5-HT- 5-hydroxytryptamine; IGF-insulin-like growth factor, SCFA, short-chain fatty acid, PTH-Parathyroid hormone.

**Figure 4 metabolites-12-01247-f004:**
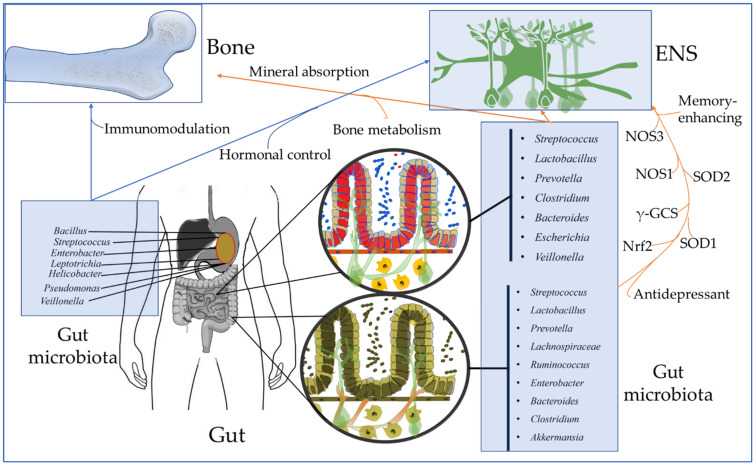
Influence of gut microbiota on enteric nervous system and bone development. The gut microbiota is known to affect the ENS via the NOS_3_, NOS_1_, SOD_2_, SOD_1_, Nrf2, γ-GCS. Gut microbiota is also known to demonstrate immunomodulation, hormonal control, and mineral absorption activities, and metabolism of bones.

**Table 1 metabolites-12-01247-t001:** Major Gut Microbiota found in the colon and their unique characteristics [52].

Bacterium	Unique Characteristics	Incidence in the Colon	Reference
*Escherichia coli*	Gram-negative, facultative anaerobic, rod-shaped, coliform bacterium	100	[53]
*Bacteroides fragilis*	Anaerobic, Gram-negative, pleomorphic to rod-shaped bacterium	100	[54]
* Prevotella melaninogenica *	Anaerobic, Gram-negative rod, with black colonies, and black pigment, not motile, non-spore forming, grows well on blood agar.	100	[55]
*Klebsiella* sp.		40–80	[56]
*Enterobacter* sp.	Rod-shaped, Gram-negative bacterium, facultative anaerobic, possesses peritrichous flagella, oxidase-negative, catalase-positive	40–80	[57]
*Bifidobacterium bifidum*	Gram-positive, anaerobic, non-motile, non- spore-forming, rod-shaped, usually in clusters, pairs, or singly.	30–70	[58]
*Staphylococcus aureus*	Gram-positive, catalase positive, nitrate positive, facultative anaerobe, spherically shaped bacterium	30–50	[59]
*Lactobacillus*	Gram-positive, aerotolerant anaerobes or microaerophilic, rod-shaped, non-spore-forming bacteria.	20–60	[60]
*Clostridium perfringens*	Gram-positive, rod-shaped, anaerobic, spore-forming bacterium	25–35	[61]
*Proteus mirabilis*	Gram negative facultative anaerobic, rod shaped, and peritrichous flagella bacterium	5–55	[62]
*Clostridium tetani*	Rod-shaped, Gram-positive, flagellated, spore forming bacterium	1–35	[63]
*Clostridium septicum*	Gram-positive, spore forming, obligate anaerobic bacterium	5–25	[64]
*Pseudomonas aeruginosa*	Encapsulated, Gram-negative, aerobic-facultatively anaerobic, rod-shaped bacterium.	3–11	[65]
*Salmonella enterica*	Rod-headed, flagellate, facultative anaerobic, Gram negative	3–7	[66]

**Table 2 metabolites-12-01247-t002:** Major gut microbes and their protective effects on bone and brain.

Gut Microbe	Effect on Bone or Brain	References
** *Lactobacillus reuteri* **	Secretes immunomodulatory factors that suppress TNF signaling and prevent bone loss. Suppresses RANKL and TRAP 5 and inhibits osteoclastogenesis	[163,179,180]
** *Bifidobacterium longum* **	Elevates osteocalcin and enhances bone formation. Reduces bone resorption parameters Upregulates Sparc and BMP-2 genes and increases bone formation Decrease hospital anxiety and depression, decreases depression and anger-hostility=	[165,181,182,183,184]
** *Lactobacillus rhamnosus* **	Decreases osteoclastogenic cytokines IL-6, IL-17 and TNF-α Increases antiosteoclastogenic cytokines IL-4, IL-10 and IFN-γ Regulates Treg-Th17 differentiation and modulates bone metabolism *Influences* emotional behavior and the expression of the neurotransmitter GABA (γ-aminobutyric acid) in the CNS in a region-dependent manner	[166,185]
** *Lactobacillus plantarum* ** **&** ** *Lactobacillus paracasei* **	Modulates bone morphogenetic proteins (BMP) Modulates RANKL pathways Alleviated abnormal expression of brain-derived neurotrophic factor (BDNF), γ-aminobutyric acid type A receptor α1 (GABA_Aα1_), γ-aminobutyric acid type B receptor1 (GABA_b1_), and 5-hydroxytryptamine receptor1A (5-HT_1A_) in the hippocampus and of GABA_Aα1_ in the prefrontal cortex. Modulates the concentrations neurotransmitters in the brain (especially dopaminergic metabolites). Reduced early-life stress abnormalities	[27,186,187,188]
** *Lactobacillus johnsonii* **	Activate bone marrow-derived dendritic cells to mature and express cytokines and chemokines in vitro Prevents psychological stress–induced memory dysfunction	[189,190]
** *Streptococcus thermophilus* **	Possess genes encoding various metabolites including gamma-aminobutyric acid producing system, crucial neurotransmitter associated with mood and stress response. Memory-enhancing	[191]
* **L. fermentum** *	Alleviates D-galactose-induced aging, activates the Nrf2 signaling pathway, increases regulatory inflammatory factors and antioxidant enzymes. Prevent aging or reduce oxidative stress. Increased serum SOD, GSH, CAT, and MDA, and decreased IL-6, IL-1β, TNF-α, and IFN-γ Upregulates the expression of Nrf2, γ-GCS, NOS1, NOS3, SOD1, SOD2, and CAT in brain tissues	[192]
* **Lactobacillus casei** *	Prevents impaired barrier function in intestinal epithelial cells As an antidepressant mediated by the microbiota-gut-brain axis.	[6,193]

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
