# Peer review of "Modulatory Effect of Gut Microbiota on the Gut-Brain, Gut-Bone Axes, and the Impact of Cannabinoids"

_metabolites, 2022, doi:10.3390/metabo12121247_

Round 1

Reviewer 1 Report

Mr. Iddrisu Ibrahim`s manuscript :Modulatory effect of gut microbiota on the gut-brain, gut-bone axes, and the impact of cannabinoidsIn this review, the dual functions gut microbiota plays in the regulation of gut-bone axis and gut-brain axis and the impact of CBD on these roles are discussed in detail.The work is well written and the text is of high quality. However, the manuscript readability can be much improved to better convey the importance of your study. With editing and some minor revisions, I feel that this manuscript will be suitable for publication.

1.Figure 2. D Del:Created in .... in figure.

2.some references lost page, e.g. reference 3,5,7,19,35.....

3.some reference,page p,pp, doi:  Please unify the format.

4. in 2.3, Can you give me an example? Give one or two examples after the summary

5. Line 315-319, 321-325, 726-733, 781-791, Need refernece.

6.line 743, Del 2022

7. add the references

1)Diet-microbiome-gut-brain nexus in acute and chronic brain injury

Maria Alexander Krakovski, Niraj Arora, Shalini Jain, Jennifer Glover, Keith Dombrowski, Beverly Hernandez, Hariom Yadav, Anand Karthik Sarma

Front Neurosci. 2022; 16: 1002266. Published online 2022 Sep 16. doi: 10.3389/fnins.2022.1002266

2)The gut microbiome as a modulator of healthy ageing

Tarini Shankar Ghosh, Fergus Shanahan, Paul W. O’Toole

Nat Rev Gastroenterol Hepatol. 2022; 19(9): 565–584. Published online 2022 Apr 25. doi: 10.1038/s41575-022-00605-x

3)Understanding the Role of the Gut Microbiome in Brain Development and Its Association With Neurodevelopmental Psychiatric Disorders

Somarani Dash, Yasir Ahmed Syed, Mojibur R. Khan

Front Cell Dev Biol. 2022; 10: 880544. Published online 2022 Apr 14. doi: 10.3389/fcell.2022.880544

Author Response

Mr. Iddrisu Ibrahim`s manuscript: Modulatory effect of gut microbiota on the gut-brain, gut-bone axes, and the impact of cannabinoids”. In this review, the dual functions gut microbiota plays in the regulation of gut-bone axis and gut-brain axis and the impact of CBD on these roles are discussed in detail.The work is well written and the text is of high quality. However, the manuscript readability can be much improved to better convey the importance of your study. With editing and some minor revisions, I feel that this manuscript will be suitable for publication.

1.Figure 2. D, Del:Created in .... in figure.

Response 1: Figure 2 has been edited to remove created in biorender.com from the figure.

2.some references lost page, e.g. reference 3,5,7,19,35

Response 2: References reformatted to suit Journal, nonetheless, there are no page numbers assigned, thus we have added the articles’ Pubmed and google scholar addresses.

3.some reference,page “p”,”pp”, “doi: “ Please unify the format.

Response 3: Page “p” has all been revised into pp. rather.

  1. in 2.3, Can you give me an example? Give one or two examples after the summary

Response 4; Section revised to include an example

  1. Line 315-319, 321-325, 726-733, 781-791, Need refernece.

Response 5: Revised to include references

6.line 743, Del “2022”

Response 6: Revised to have “2022” removed.

  1. add the references

1)Diet-microbiome-gut-brain nexus in acute and chronic brain injury

Maria Alexander Krakovski, Niraj Arora, Shalini Jain, Jennifer Glover, Keith Dombrowski, Beverly Hernandez, Hariom Yadav, Anand Karthik Sarma

Front Neurosci. 2022; 16: 1002266. Published online 2022 Sep 16. doi: 10.3389/fnins.2022.1002266

2)The gut microbiome as a modulator of healthy ageing

Tarini Shankar Ghosh, Fergus Shanahan, Paul W. O’Toole

Nat Rev Gastroenterol Hepatol. 2022; 19(9): 565–584. Published online 2022 Apr 25. doi: 10.1038/s41575-022-00605-x

3)Understanding the Role of the Gut Microbiome in Brain Development and Its Association With Neurodevelopmental Psychiatric Disorders

Somarani Dash, Yasir Ahmed Syed, Mojibur R. Khan

Front Cell Dev Biol. 2022; 10: 880544. Published online 2022 Apr 14. doi: 10.3389/fcell.2022.880544

Response 7: All references were used to update the manuscript. Thank you. 

Reviewer 2 Report

I read the manuscript sent for evaluation with interest. It deals with interesting topics related to the influence of gut microflora on the functioning of the human body. I am not sure if these genes should have been combined with canabinoids - but this is the authors' decision.

I have a few comments to the text:
1. You have characterized CBD and THC very nicely. However, from my point of view - a reader who deals with microbiology - I missed a very important aspect: the characteristics of the microorganisms that make up the gut microbiome. I propose to add a subsection (preferably at the beginning of the article), in which the Authors will list the most characteristic microorganisms and describe features other than whether they are Gram-positive or Gram-negative (this information was), their role in gut or population size of given types/species in a specific part of the gut.

2. Do you like the words "2.0", "3.0" or 7.0 "? It looks very unnatural, so please remove these zeros and leave the main chapters with an integer without the decimal point.

3. Literature cited in references: standardize the entries, because sometimes Authors use the abbreviation of the journal name, sometimes it is the full name of the journal. It often happens that names are not italicized and you use lowercase letters in many parts of the name - usually they are all in uppercase. The list must be prepared in accordance with the requirements of the Editorial Board and please adapt it to these requirements.

4. English is not my mother tongue, but I don't think the sentence "The gut is composed of a dynamic organ that ..." is correct. It is a scientific article and should be precise in reception. There were several such sentences in the text.

Author Response

Response 1: Sections added to give few examples of microbiota and some tabulated characteristics.

  1. Do you like the words "2.0", "3.0" or 7.0 "? It looks very unnatural, so please remove these zeros and leave the main chapters with an integer without the decimal point.

Response 2. Numbering style changed to contain no zeros at the end

  1. Literature cited in references: standardize the entries, because sometimes Authors use the abbreviation of the journal name, sometimes it is the full name of the journal. It often happens that names are not italicized and you use lowercase letters in many parts of the name - usually they are all in uppercase. The list must be prepared in accordance with the requirements of the Editorial Board and please adapt it to these requirements.

Response 3: References worked on to standard form

  1. English is not my mother tongue, but I don't think the sentence "The gut is composed of a dynamic organ that ..." is correct. It is a scientific article and should be precise in reception. There were several such sentences in the text.

Response 4. Complex sentences have been rewritten into simple straightforward English

Reviewer 3 Report

The article concerns a number of sharply discussed relationships between the human host and the gut microbiota in view of its vital significance for the basic body functions. In particular, the authors consider sufficient modifying effects of cannabinoids (THC and CBD), especially in neuroinflammatory disorders. Gut-brain, gut-bone, gut-lung, and gut-heart interactions are also discussed, as well as the role of gut microbiome in cancer, bone remodeling, etc.

The remarks and questions concern, mainly, general design and structure of the review. E.g., the authors discuss gut-brain axis, as well brain-bone interactions and hormonal effects, however, they do it in descriptive matter, without definite generalization, thus making their conclusions less convincing. (Section 2.2., pages 6-7). The sections on homeostatic, non-homeostatic  and obesity-related physiopathology seem to be only loosely associated.    Of sufficient interest are the data on gut microbiome-brain regulation, with next sections 11 and 12 which concerning effects of cannabinoids and their interactions with gut microbiome functions.    The second part of review seems to be most interesting, beginning from 6.1 (page 11).  The authors do not, however, suggest the main lines of interactions between cannabinoids and gut microbiome-brain axis. The first part of review contains, generally, known data and references on the matter.

Abbreviations (e.g., THC and CBD) should be deciffered just from Summary (later, lines 81, 90)

In the long text of conclusion (page 24), all the main axiomas concerning gut microbiome and its interactions are listed, however, without mentioning the potential effects of cannabinoids as presumed in the first section. Hence, the conclusion should be consolidated, in view of the complex effects of cannabinoids on the gut-brain axis (may be, supplied with an understandable figure). Therefore, the ultimate goal of this review formulated at the end of Introduction was not achieved, as seen from Conclusion.

Moreover, some passages are hard to understand, e.g.,: Line 71: Medicinal cannabis has derived the taste and interest of many people for a long time …, and some other misprints throughout the text.

The article may be published with clear conclusion on the declared interference of cannabinoids with functioning of gut-brain axis.

Sufficient copy-editing is strongly recommended.

Author Response

The article concerns a number of sharply discussed relationships between the human host and the gut microbiota in view of its vital significance for the basic body functions. In particular, the authors consider sufficient modifying effects of cannabinoids (THC and CBD), especially in neuroinflammatory disorders. Gut-brain, gut-bone, gut-lung, and gut-heart interactions are also discussed, as well as the role of gut microbiome in cancer, bone remodeling, etc.

The remarks and questions concern, mainly, general design and structure of the review. E.g., the authors discuss gut-brain axis, as well brain-bone interactions and hormonal effects, however, they do it in descriptive matter, without definite generalization, thus making their conclusions less convincing. (Section 2.2., pages 6-7). The sections on homeostatic, non-homeostatic and obesity-related physiopathology seem to be only loosely associated.    Of sufficient interest are the data on gut microbiome-brain regulation, with next sections 11 and 12 which concerning effects of cannabinoids and their interactions with gut microbiome functions.    The second part of review seems to be most interesting, beginning from 6.1 (page 11).  The authors do not, however, suggest the main lines of interactions between cannabinoids and gut microbiome-brain axis. The first part of review contains, generally, known data and references on the matter.

Responses

Abbreviations (e.g., THC and CBD) should be deciffered just from Summary (later, lines 81, 90)

Response: Abbreviation for THC and CBD has been deciphered in the abstract.

In the long text of conclusion (page 24), all the main axiomas concerning gut microbiome and its interactions are listed, however, without mentioning the potential effects of cannabinoids as presumed in the first section. Hence, the conclusion should be consolidated, in view of the complex effects of cannabinoids on the gut-brain axis (may be, supplied with an understandable figure). Therefore, the ultimate goal of this review formulated at the end of Introduction was not achieved, as seen from Conclusion.

Response: The conclusion has been revised most part of the manuscript been rewritten to improve understanding

Moreover, some passages are hard to understand, e.g., Line 71: Medicinal cannabis has derived the taste and interest of many people for a long time …, and some other misprints throughout the text.

Response: This line has been rephrased, and similar lines has been rewritten.

The article may be published with clear conclusion on the declared interference of cannabinoids with functioning of gut-brain axis.

Response: The conclusion has been revised to include briefly the impact of THC and CBD on gut microbiota

Sufficient copy-editing is strongly recommended.

Response: Most part of the manuscript has been revised.

Reviewer 4 Report

Authors have chosen a very good topic and timely review on the role of gut microbiota on the gut-brain, gut-bone axes, and the impact of cannabinoids. The comprehensive coverage of scientific literature could have made this article quite interesting. However, I believe the lead author doesnt have experience of writing a quality Review article. They have mentioned the results of other studies in the abstract section which should cover a brief introduction, requirement of this article and their intent to cover certain topic. Hence, I must reject this paper at current state

Author Response

Authors have chosen a very good topic and timely review on the role of gut microbiota on the gut-brain, gut-bone axes, and the impact of cannabinoids. The comprehensive coverage of scientific literature could have made this article quite interesting. However, I believe the lead author doesn’t have experience of writing a quality Review article. They have mentioned the results of other studies in the abstract section which should cover a brief introduction, requirement of this article and their intent to cover certain topic. Hence, I must reject this paper at current state.

Response: Thank you very much for your readership. We appreciate your view.

Round 2

Reviewer 1 Report

Thank you very much!

Author Response

Response:

Our sincere thanks to the reviewer for the thorough revision of the manuscript. We appreciate his/her time, effort and scholarship. The reviewer’s comments have tremendously improved our manuscript, and we appreciate it a lot.

Thank you and have a wonderful Weekend.

Reviewer 3 Report

The article was recommended for careful revision, due to mangling data on cannabinoid effects upon intestinal microbiome and gut/brain axis, and absence of clear conclusions.

The revised version is supplemented by abundant data on the main bacterial phyla in different gut compartments (Sections 1.1 to 1.4, Table 1).  These data, generally, have no direct connection with subsequent discussion of cannabinoid effects, and take much place. The authors should be concentrated on cannabinoid effects and appropriate microbiota and neurobiological changes, as declared in the title and Summary.

The authors have also added some data on gastrointestinal effects of cannabinoids.

The list of references is now extended to 262 citations (ca. 200 in the 1st version).

 Some added sentences still are stylish wrong, or are hard to understand, for example, … rise of mechanized occupations and the transition from more labor intensive to sedentary work systems…(lanes 270-271).

The term hypothalamic pituitary adrenal axis (lanes 447-448) is repeated twice in the same sentence, thus requiring abbreviation, or deletion of this term.

 Anyway, the article needs careful language editing.

 In general, the revised version of this article now better meets the goals declared. It contains more information on potential cannabinoid effects on brain/gut axis (lanes 805-806), and in neuroinflammation (lanes 857-867), but these studies and findings are rather scarce, as seen from the existing literature.

In summary, this review article could be published, however, after extensive copy-editing with respect to English grammar and style.  

Author Response

The article was recommended for careful revision, due to mangling data on cannabinoid effects upon intestinal microbiome and gut/brain axis, and absence of clear conclusions.

Response: Conclusion has been extensively revised.

The revised version is supplemented by abundant data on the main bacterial phyla in different gut compartments (Sections 1.1 to 1.4, Table 1).  These data, generally, have no direct connection with subsequent discussion of cannabinoid effects, and take much place. The authors should be concentrated on cannabinoid effects and appropriate microbiota and neurobiological changes, as declared in the title and Summary.

Response: The manuscript has been revised to include more of cannabinoid effects.

The authors have also added some data on gastrointestinal effects of cannabinoids.

The list of references is now extended to 262 citations (ca. 200 in the 1st version).

 Some added sentences still are stylish wrong, or are hard to understand, for example, … rise of mechanized occupations and the transition from more labor intensive to sedentary work systems…(lanes 270-271).

Response: Sentences have been revised

The term hypothalamic pituitary adrenal axis (lanes 447-448) is repeated twice in the same sentence, thus requiring abbreviation, or deletion of this term.

Response: hypothalamic pituitary adrenal axis has been deleted.

 Anyway, the article needs careful language editing.

Response: Extensive language editing has been carried out.

 In general, the revised version of this article now better meets the goals declared. It contains more information on potential cannabinoid effects on brain/gut axis (lanes 805-806), and in neuroinflammation (lanes 857-867), but these studies and findings are rather scarce, as seen from the existing literature.

Response: This section has been extensively edited to contain any information we have obtained further during this revision.

In summary, this review article could be published, however, after extensive copy-editing with respect to English grammar and style.  

Response: Extensive copy-editing has been carried out.

Reviewer 4 Report

I didn't see any change made by the author in response to my earlier comments. I stand by my earlier recommendation

Author Response

Response:

Our sincere thanks to the reviewer for his/her opinion concerning the manuscript, we appreciate it.

Thank you and have a wonderful Weekend.

Summary has been revised, the entire document has been completely revised. Abstract has been edited. More information about Cannabinoid has been added.
